# Overshoot dependence on the cross-shock potential

Michael Gedalin[1], Xiaoyan Zhou[2], Christopher T. Russell[2], and Vassilis Angelopoulos[2]

[1]Department of Physics, Ben Gurion University of the Negev, Beer-Sheva, Israel
[2]Department of Earth, Planetary, and Space Sciences, University of California, Los Angeles, USA

**Correspondence:** gedalin@bgu.ac.il

**Abstract.** Coherent downstream oscillations of the magnetic field in shocks are produced due to the coherent ion gyration and quasi-periodic variations of the ion pressure. The amplitude and the positions of the pressure maxima and minima depend on the cross-shock potential and upstream ion temperature. Two critical cross-shock potentials are defined: the critical gyration potential (CGP) which separates the cases of increase or decrease of the component of the velocity of the distribution center along the shock normal, and the critical reflection potential (CRP) above which ion reflection becomes significant. In weak very low upstream kinetic-to-magnetic pressure ratio, $\beta$, shocks CRP exceeds CGP. For potentials below CGP the first downstream maximum of the magnetic field is shifted farther downstream and is larger than the second one. For higher potentials the first maximum occurs just behind the ramp and is lower than the second one. With the increase of the upstream temperature CGP exceeds the CRP. For potentials below CRP the effects of ion reflection are negligible and the shock profile is similar to that of very low $\beta$ shocks. If the potential exceeds CRP ion reflection is significant, the magnetic field increase toward the overshoot becomes steeper, and the largest peak occurs at the downstream edge of the ramp.

*Copyright statement.* TEXT

## 1 Introduction

Collisionless shocks (CS) are one of the most efficient accelerators of charged particles in the Universe. They are present in virtually all plasma environments at the scales from $\sim 1$ cm in the terrestrial labs to $\sim 1$ Mpc in galaxy clusters. CS is a multi-scale object, where highest energies are achieved at largest scales within the diffusive process due to scattering at fluctuations far upstream and far downstream and multiple crossings of the shock. The latter occur within the scatter-free region so that ion dynamics in the shock front is intimately related to the large scale acceleration: while the diffusive acceleration occurs at scales much larger than the shock width, the spectrum of the accelerated particles is essentially determined by conservation laws at the scatter-free shock transition. The fields in the shock front are responsible for ion heating, generation of backstreaming ion beams (Burgess, 1987; Kucharek et al., 2004; Oka et al., 2005; Gedalin et al., 2008; Gedalin, 2016b), acceleration of pickup ions (Lee et al., 1996; Zank et al., 1996; Zilbersher and Gedalin, 1997; Ariad and Gedalin, 2013), and injection into the diffusive mechanism (Scholer et al., 2002; Giacalone, 2005). Thus, the structure of the shock front is the central problem of the shock physics. The shock structure can be studied within in situ measurements only at heliospheric shocks. Qualitative

understanding of the shock structure substantially improved due to these high quality observations and also due to numerical simulations. The frontier of the observational shock studies has shifted recently towards the processes occurring within few ion convective gyroradii in both directions from the ramp along the shock normal (Dimmock et al., 2012; Wilson et al., 2012, 2014; Johlander et al., 2016; Burgess et al., 2016; Eselevich et al., 2017; Wilson III et al., 2017; Gingell et al., 2017).

Magnetic profiles of collisionless shocks are rarely monotonic, even for low-Mach numbers (Greenstadt et al., 1975; Greenstadt et al., 1980; Russell et al., 1982a; Mellott and Greenstadt, 1984; Farris et al., 1993; Balikhin et al., 2008; Russell et al., 2009; Kajdič et al., 2012). Since the peak value of the downstream oscillations increases with the increase of the Mach number, for a long time is was believed that overshoots are produced by ion reflection in super-critical shocks (Livesey et al., 1982; Russell et al., 1982b; Sckopke et al., 1983; Scudder et al., 1986; Mellott and Livesey, 1987). Super-critical shocks are

the shocks with the Mach number exceeding the critical Mach number (Edmiston and Kennel, 1984; Kennel, 1987), so that resistivity (Edmiston and Kennel, 1984) and thermal conductivity (Kennel, 1987) alone cannot provide necessary dissipation to sustain a shock. Eventually coherent downstream oscillations were observed at a very low-Mach number shock (Balikhin et al., 2008) with the Alfvenic Mach number of $M = 1.3$ and magnetic compression of $B_d/B_u = 1.3$. The oscillating trail behind the ramp exhibited all features expected for a supercritical shocks, like the largest first peak, spatially periodical peaks, and gradual

decrease of the peak amplitude. Such oscillations, albeit often less ordered, were found to be common in low-Mach number shocks (Russell et al., 2009; Kajdič et al., 2012). They were successfully explained as a result of coherent ion gyration upon crossing the shock ramp and subsequent collisionless relaxation due to gyrophase mixing (Balikhin et al., 2008; Ofman et al., 2009; Ofman and Gedalin, 2013; Gedalin, 2015; Gedalin et al., 2015, 2018). It has been shown that the largest peak amplitude is determined mainly by the magnetic compression and cross-shock potential, while the damping rate of the oscillations is

related to the upstream thermal-to-fluid speed ratio (Gedalin, 2015). Shapes of the downstream profile, like relative peaks of the first oscillations and steepness of the magnetic field increase up to the first peak, vary considerably among observed shocks, even subcritical ones. Sufficient attention has not been devoted so far to the relation of the details of the magnetic oscillation pattern to the shock parameters and ion kinetics in the shock front. In particular, amplitudes and positions of the first peaks, which are not yet distorted by gyrophase mixing, may provide information about the cross-shock potential as well about the

ion transmission and reflection.

## 2   Weak low-$\beta$ shocks

In what follows $B_u$ is the upstream magnetic field magnitude, $T_u$ is the upstream ion temperature, $n_u$ is the upstream ion number density, $v_T = \sqrt{T_u/m}$ is the upstream ion thermal speed, $m$ is the ion mass, and $\beta = 8\pi n_u T_u/B_u^2$ is the upstream kinetic-to-magnetic pressure ratio. The corresponding parameters for electrons are denoted by adding index $e$. In order to

explain the basic mechanism of producing the downstream oscillations, let us consider a simplified model of a perpendicular shock. We treat the shock as a jump in the magnetic field from $B_u$ to $B_d = RB_u$ occurring within a narrow ramp. Accordingly, the fluid drift speeds upstream and downstream are $V_u$ and $V_d = V_u/R$. We shall also neglect the electron contribution in the plasma pressure and treat ions as a monoenergetic beam entering the shock with the velocity $V_u$ along the shock normal. The

analysis is done in the normal incidence frame, where $x$ is along the shock normal (toward downstream) and $z$ is along the magnetic field. The equations of motion for ions inside the ramp are

$$\dot{v}_x = \frac{q}{m}E_x + \frac{q}{mc}v_yB_z \tag{1}$$

$$\dot{v}_y = \frac{q}{mc}(V_uB_u - v_xB_z) \tag{2}$$

We integrate the equations of motion across the ramp assuming $|v_y| \sim v_T \ll V_u$, where $v_T$ is the thermal speed of upstream ions. In this approximation we get

$$\frac{u_x^2}{2} - \frac{V_u^2}{2} = -\frac{q}{m}\phi + \int \frac{q}{mc}v_yB_z dx \tag{3}$$

$$u_y = \int \frac{qB_u}{mc}\left(\frac{V_u}{v_x} - \frac{B_z}{B_u}\right) dx \tag{4}$$

Here $u$ denotes the ion velocity at the downstream edge of the ramp while $v(x)$ denotes the ion velocity at the position $x$ inside the ramp. The second term in (3) is a small correction for ramp width $\lesssim (c/\omega_{pi})$ and $v_T/V_u \ll 1$. Here $(c/\omega_{pi})$ is the ion inertial length. This small correction can be neglected for our purposes. In (4) the only term is small but nonzero. Thus, if the cross-shock potential is $\phi = s(mV_u^2/2e)$, the ion velocity just after crossing the jump is

$$v_x(x=0) = V_u\sqrt{1-s}, \quad v_y(x=0) = u_y \tag{5}$$

The ion motion is then described as a drift along the shock normal with the velocity $V_u/R$ and gyration around the magnetic field:

$$v_x(t) = V_d + v_\perp \cos(\Omega_d t + \varphi) \tag{6}$$

$$v_y(t) = v_\perp \sin(\Omega_d t + \varphi) \tag{7}$$

$$v_\perp^2 = (V_u\sqrt{1-s} - R)^2 + u_y^2 \tag{8}$$

where $\Omega_d = eB_d/m_ic$ is the downstream ion gyrofrequency. For a cold beam all ions move together and the coordinate along the shock normal is given by

$$x(t) = V_d t + \frac{v_\perp}{\Omega}[\sin(\Omega_d t + \varphi) - \sin\varphi] \tag{9}$$

In general, it is not possible to derive an analytical expression for $v_x(x)$. For our purposes it is sufficient to restrict ourselves to weak gyration, $v_\perp < V_d$, so that $dx(t)/dt = v_x > 0$ and $x(t)$ is invertible, that is, $t(x)$ is a single-valued function. Let us define the critical gyration potential (CGP) $s_{cr} = 1 - 1/R^2$. For $s < s_{cr}$ the initial gyrophase $\varphi \approx 0$, so that $dv_x/dx < 0$ at the downstream edge of the ramp. For $s > s_{cr}$ the initial gyrophase is $\varphi \approx \pi$, so that $dv_x/dx > 0$ at the downstream edge of the ramp.

The total (dynamic and kinetic) ion pressure is given by

$$p_{i,xx} = m_i n v_x^2 = m_i n_u V_u v_x \tag{10}$$

where we have used the mass conservation $nv_x = n_u V_u$. Pressure balance requires $p_{i,xx} + B^2/8\pi = \text{const}$, so that the magnetic field has maxima at the minima of the ion pressure. The latter occur at the minima of $v_x$. For $s < s_{cr}$ the velocity decreases inside the ramp and keeps decreasing down to $v_{x,min} = V_d - v_\perp$ at $\Omega_d t + \varphi = \pi$ which approximately corresponds to $x_l = \pi V_d/\Omega_d$ for $\varphi \approx 0$. Thus, the first maximum of the magnetic field occurs at $x_l$ at the pressure $p_l = m_i n_u V_u (V_d - v_\perp)$. With the increase of $s$ the relative contribution of $u_y$ in $v_\perp$ increases which moves the position of the first pressure minimum closer to the ramp. For $s > s_{cr}$ the velocity decreases inside the ramp but starts to increase just behind it. Thus, the first maximum of the magnetic field occurs at $x = 0$ (the downstream edge of the ramp) at the pressure $p_h = m_i n_u V_u (V_d - u_x)$. Since $u_x < v_\perp$ one has $p_h > p_l$ which means that the first peak will be lower than the subsequent ones corresponding to the pressure minima $p_l$.

For a cold ion beam the amplitude of further pressure oscillations does not change. Finite temperature leads to the divergence of the ion trajectories and gradual gyrophase mixing. The divergence occurs already at the shock crossing since the downstream ion velocity $v_{x,d} = \sqrt{v_{x,u}^2 - 2e\phi/m}$, and the spread in $v_{x,u}$ results in a more substantial spread in $v_{x,d}$. Moreover, there is nonzero $v_y$ which affects the gyration speed $v_\perp$ and $\varphi$, which are now different for different particles:

$$v_\perp^2 = \left( \sqrt{v_{x,u}^2 - 2e\phi/m} - V_d \right)^2 + v_{y,u}^2 \tag{11}$$

$$\cos\varphi = \frac{\sqrt{v_{x,u}^2 - 2e\phi/m} - V_d}{v_\perp} \tag{12}$$

The downstream ion pressure including finite temperature is obtained as an integral over the distribution

$$p_{i,xx} = m_i \int v_x^2 f(\boldsymbol{v}) d^3\boldsymbol{v} \tag{13}$$

It has been shown (Gedalin et al., 2015; Gedalin, 2016a) that finite temperature results in the collisionless relaxation during which the downstream ion distribution gyrotropizes and the pressure oscillations damp out. The relaxation is faster for larger $v_T/V_u$. In oblique shocks the mechanism of the generation of downstream oscillations is the same. Relaxation is faster for lower angles $\theta$ between the shock normal and the upstream magnetic field (Gedalin, 2015; Gedalin et al., 2015).

With the increase of the magnetic compression CGP rapidly increases. At $R = 2$ this critical value is $s_{cr} = 0.75$. Although such high cross-shock potentials cannot be completely excluded, they are not observed often (Dimmock et al., 2012). Thus, we expect that in most shocks the potential is below CGP. Yet, in many shocks the first magnetic peak occurs right at the downstream edge of the ramp. In many cases it is also the largest peak. The above analysis is valid, strictly speaking, only for sufficiently low-$\beta = 8\pi n_u T_u/B_u^2$ shocks since the number of quasi-reflected and/or reflected ions rapidly increases with the increase of $v_T/V_u$, where $v_T = \sqrt{T_u/m}$ is the upstream thermal speed of ions (Gedalin, 2016b). In the narrow shock approximation all ions having initially $mv_x^2/2 < e\phi$ cannot cross the ramp. This mode of reflection is efficient when $1 - \sqrt{s} \sim v_T/V_u$. Deceleration of quasi-reflected ions inside the ramp can be expected to result in faster reduction of the ion pressure with the distance from the upstream edge of the ramp, that is, steeper increase of the magnetic field.

## 2.1 Advanced test particle analysis vs observations

The principles of the advanced test particle analysis have been described in detail by Gedalin and Dröge (2013). In brief, a model magnetic field profile is chosen, supplemented with a model electric field shape. The basic upstream plasma parameters, that is, ion and electron $\beta$ and the angle between the shock normal and the upstream magnetic field $\theta$ are chosen and remain fixed during the analysis. Choosing a magnetic compression ratio $R$, the rest of the significant parameters are varied. With each set of the parameters ions are numerically traced across the shock, the ion pressure is determined, and the corresponding magnetic field is derived from the pressure balance. The parameters are varied until reasonable agreement is achieved with the adopted model profile: the asymptotic values of the magnetic field should be equal and the fluctuations as small as possible. It has been found that the most influential parameters are the Alfvenic Mach number $M$ and the normalized cross-shock potential $s$. There is also weak dependence on the shock width $D$. The magnetic profile chosen for the analysis is taken in the following form:

$$B_z = B_u \sin\theta \left[ 1 + \frac{R-1}{2} \left( 1 + \tanh\frac{3x}{D} \right) \right] \tag{14}$$

with $B_x = B_u \cos\theta$, $B_y \propto dB_z/dx$, and $E_x \propto dB_z/dx$. The coefficients of proportionality are constrained by the chosen values of the normal incidence frame cross-shock potential $s_{NIF}$ and the de Hoffman-Teller potential $s_{HT}$ (Goodrich and Scudder, 1984; Scudder et al., 1986; Schwartz et al., 1988). The latter was found to almost not affect the ion motion and was kept $s_{HT} = 0.1$ in the subsequent analysis. The post-tracing magnetic field was derived from the condition

$$p_e + p_{i,xx} + \frac{B^2}{8\pi} = \text{const} \tag{15}$$

where the ion pressure was determined numerically and for the electron pressure the polytropic equation of state $p_e/n^{5/3}$ was used, together with the quasineutrality.

Figure 1 shows the results of the numerical analysis for both high (top) and low (bottom) potentials. In both cases the magnetic compression $R = 1.45$, the Alfvenic Mach number $M = 1.4$, the shock angle $\theta = 70°$, the upstream $\beta_i = \beta_e = 0.05$, and the width $D = r_g/M$ are the same. CGP is $s_{cr} = 1 - 1/R^2 \approx 0.52$ in this case. It appears that the chosen shock parameters allow two different cross-shock potential values. The positions of the first two peaks and their values are shown for convenience. The coordinate is measured in $r_g = V_u/\Omega_u$. It is clearly seen that for the low potential the first peak is shifted farther downstream from the ramp and its amplitude is higher than that of the second peak. In the case of the higher potential the first peak occurs at the downstream edge of the ramp and its amplitude is lower than that of the second one. Figure 2 illustrates the difference in the behavior of the normal component of the ion velocity, $v_x$, in both cases. In the low potential case this component continues to decrease well beyond the ramp. Subsequent dips become more and more shallow with the distance from the ramp. In the high potential case $v_x$ starts to increase upon crossing the ramp. The second dip is deeper because lower $v_x$ are achieved, as explained above.

Parameters of the above analysis have been chosen close to those for two THEMIS-C crossings of the Earth bow shock, 2011-03-30/08:09:40 and 2011-03-30/08:51:40 (Pope et al., 2019). The magnetic profiles for these crossings are shown in

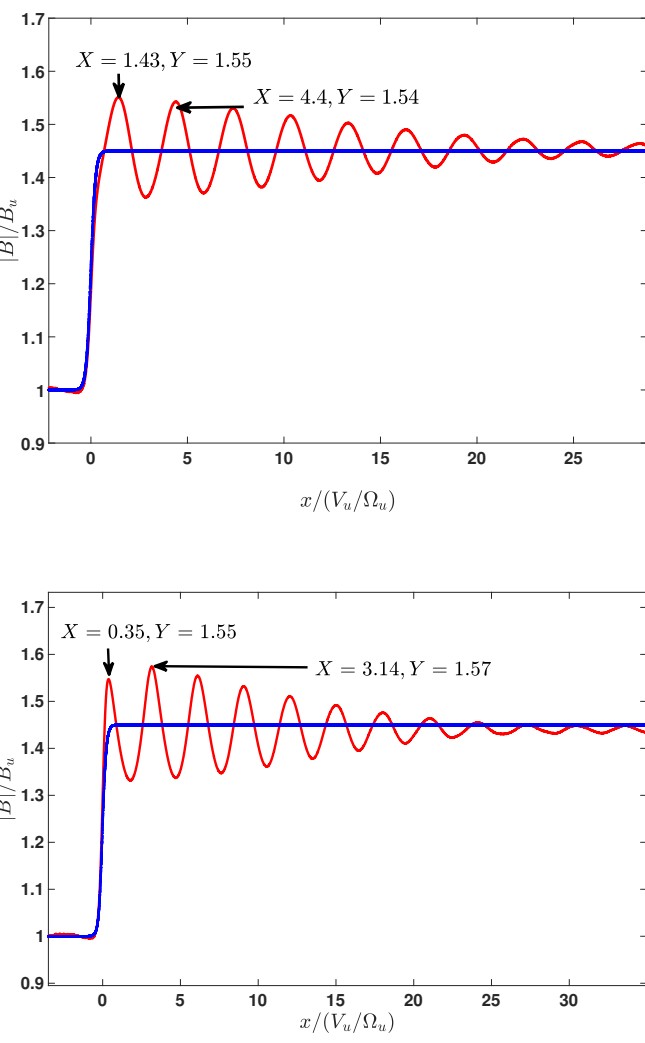

**Figure 1.** Comparison of the derived magnetic profiles (red) for $s_{NIF} = 0.4$ (top panel) and $s_{NIF} = 0.65$ (bottom panel). In both cases $R = 1.45$, $M = 1.4$, and $D = 1/M$. The model magnetic field is shown by blue line.

Figure 3 together with the ion spectrogram. The anti-correlation of the magnetic magnitude and the downstream ion pressure (greenish areas) are seen quite clearly at both shocks. Cross-shock potentials were calculated directly from observations (Pope et al., 2019) and found to be $s = 0.36$ for the left panel shock and $s = 0.50$ for the right panel shock. CGP is $c_{cr} \approx 0.4$ for both shocks. Thus, the magnetic field profile of the shock in the left panel of Figure 3 can be expected to be similar to that of the top panel of Figure 1 while the right panel observed shock should be similar to the bottom panel model shock. Indeed,

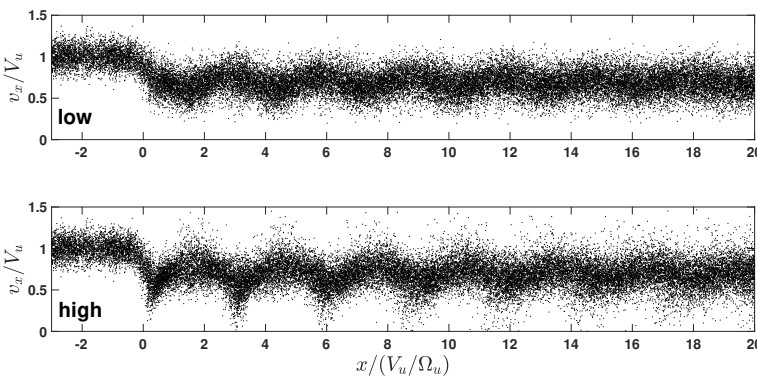

**Figure 2.** Velocity $v_x$ of ions for the low (top) and high (bottom) potentials.

the positions and relative amplitudes of the first magnetic peaks in the observed shocks are in excellent agreement with the theoretical predictions.

With the increase of the magnetic compression CGP rapidly increases. For $B_d/B_u = 2$ CGP is rather high: $c_{cr} = 0.75$. In most shocks the cross-shock potential is expected to be below this value (Dimmock et al., 2012). In low-$\beta_i$ plasmas all ions are directly transmitted across the shock without reflection and the above findings can be summarized as follows: a) below CGP the first peak is the strongest, b) with the increase of the potential toward CGP the first peak moves closer to the ramp, c) upon crossing CGP the first peak stands at the downstream edge of the ramp and is no longer the strongest.

## 3 Effects of ion reflection

Ion reflection occurs in supercritical and marginally critical shocks. Ion reflection is a kinetic process and the fate of an ion entering a shock front depends on the initial velocity of the ion. There are two major modes of ion reflection: post-ramp and in-ramp reflection. Post-ramp reflection occurs when an ion crosses the ramp, gyrates behind it, and returns back to the ramp to cross it toward upstream, but turns around again inside the ramp moving toward downstream. In-ramp reflection occurs when an ion changes its direction of motion inside the ramp and starts moving toward upstream. In both modes reflection occurs due to the combined effects of the electric and magnetic forces. Since the transition from upstream to ramp and further downstream is continuous, there is no strict separation between the two modes. Efficiency of the post-ramp reflection increases most strongly with the increase of the magnetic compression $B_d/B_u$. It also increases with the increase of the ratio $v_T/V_u = \sqrt{\beta_i/2}/M$ and with the *decrease* of the cross-shock potential $s$ (Gedalin, 1996). The inverse dependence on the cross-shock potential is related to the fact that chances of a downstream gyrating ion to return to the ramp are higher if the gyration speed is higher, while the cross-shock potential takes energy from an ion upon crossing the ramp. Efficiency of in-ramp reflection increases with the increase of the ratio $v_T/V_u$ and the cross-shock potential $s$ (Gedalin et al., 2008; Gedalin, 2016b). It can be most simply explained in the approximation of specular reflection which ignores magnetic deflection. A particles

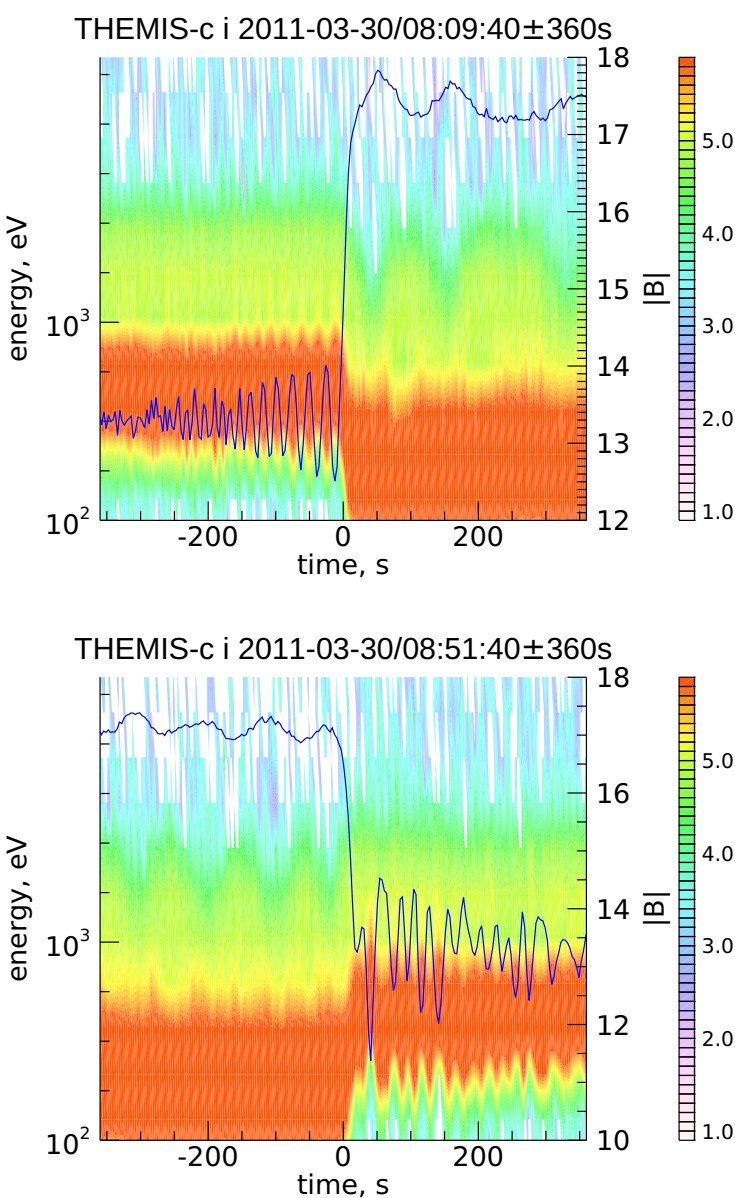

**Figure 3.** Earth bow shock crossings by THEMIS-C on 2011-03-30. Magnetic profiles (magnitude) and ion spectrograms are plotted together.

with initial $v_x$ is reflected within the ramp if $m_i v_x^2 / 2 < q\phi$. For an initial Maxwellian distribution, 5% of incident ions are reflected if $m_i(V_u - 2v_T)^2 / 2 = q\phi$ which allows us to define the critical reflection potential (CRP) $s_{5\%} = (1 - v_T / V_u)^2$. In this approximation in-ramp reflection does not depend either on the magnetic compression or shock angle and is stronger for

lower Mach numbers for given $\beta_i$ and $s$. In reality, magnetic deflection enhances the reflection which is never specular. In what follows we distinguish between reflected and quasi-reflected ions. Figure 4 illustrates the difference between ion populations and the terminology proposed by Gedalin (2016b). The first turning point is the first point along the ion trajectory where the

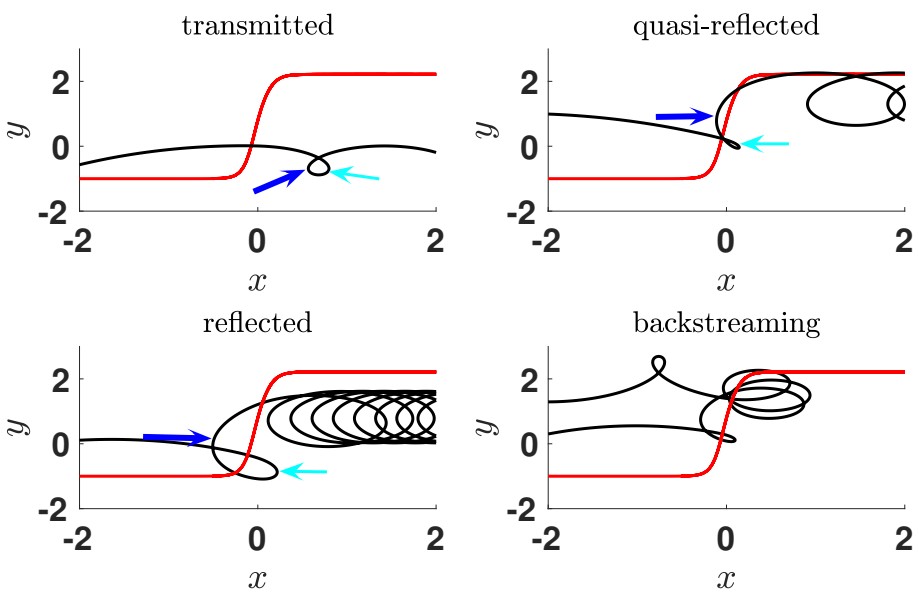

**Figure 4.** Trajectories $x - y$ of various kinds. The magnetic field shape (not to scale) is shown by red line. Cyan arrow: first turning point, blue arrow: second turning point.

sign of $v_x$ changes from positive (toward downstream) to negative (toward upstream). The second turning point is the first point
at the ion trajectory where $v_x$ changes its sign from negative to positive. A directly transmitted ion may have no turning points at all and may have turning points behind the ramp. Figure 4, top left panel, shows a trajectory of a directly transmitted ion which does have turning points. In this case the second turning point, marked with a blue arrow, occurs behind the ramp. The ion trajectory shown on the top right panel belongs to a quasi-reflected ion. In this case the second turning point occurs inside the ramp. For a reflected ion (bottom left panel) the second turning point is in the upstream region ahead of the ramp. Bottom right
panel shows, for completeness, a trajectory of a backstreaming ion which has several turning points in the ramp vicinity and eventually escapes toward upstream. Quasi-reflected and reflected ions have similar energies and similar gyrating distributions. The difference is that quasi-reflected ions do not appear in the upstream region and do not contribute to foot formation. Each reflected or quasi-reflected ion makes a loop and moves along the shock front. As a result, all these ions acquire energy in NIF so that they should be clearly distinguished from the directly transmitted ions inside the ramp and behind it, both in a
distribution plot or in a spectrogram. In both cases there should be a noticeable gap between the two.

    In low-$\beta_i$ and small $B_d/B_u$ both modes of reflection should be suppressed. In high Mach number shocks $B_d/B_u$ is large while $v_T/V_u = \sqrt{\beta_i/2}/M$ is small unless $\beta_i$ is large. In such shocks post-ramp reflection should dominate. In marginally critical and weakly supercritical shocks in-ramp reflection should dominate unless $\beta_i$ is too small. One can expect that in-ramp

reflection would cause a sharper drop of the ion pressure and therefore a steeper increase of the magnetic field. A more detailed analysis can be done numerically where the cross-shock potential $s$ and ion $\beta_i$ are fully controlled.

Figure 5 shows the results of the test-particle adjustment for a shock with $\beta_i = 0.2$ and magnetic compression $R = 1.85$.

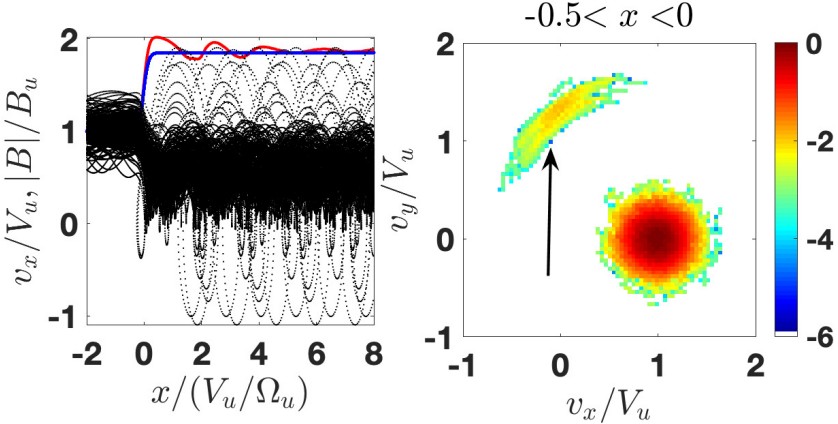

**Figure 5.** Left panel: the model (blue) and the predicted (red) magnetic profiles and the ion orbits $x$ vs $v_x$. Right panel: a slice of ion distribution inside the ramp. The shock parameters are $M = 2.1$, $\theta = 65°$, $R = 1.85$, $\beta_i = 0.2$, $\beta_e = 0.35$, and $s = 0.65$. The arrow points to the (quasi)reflected population.

The adjustment of the downstream magnetic field predicted by the test-particle analysis to the initial model field is achieved
with the cross-shock potential $s = 0.65$, which is below the corresponding CGP $s_{cr} = 0.7$ but above the corresponding CRP $s_{5\%} = 0.49$. The profile (left panel) shows a steeper increase toward the overshoot with the first peak exceeding the subsequent peaks. The same panel shows the ion orbits and the right panel shows a slice of the ion distribution which covers a half of the ramp adjacent to the upstream. Both clearly show the presence of a non-gyrotropic distribution of quasi-reflected ions. The incident and quasi-reflected populations are clearly separated in the velocity space and in energies.

Figure 6 shows the results of the test-particle adjustment for the same compression ratio and cross-shock potential but lower $\beta_i = 0.05$. In this case there are very few quasi-reflected ions and the shock profile follows the low-$\beta$ prescription shown in the top panel of Figure 2. The magnetic field increase toward the overshoot is less steep and the first peak is shifted further downstream.

Figure 7 shows the results of the test particle adjustment for the same compression ratio and $\beta_i = 0.2$ but lower cross-shock
potential $s = 0.4$. In this case there are also very few quasi-reflected ions and the shock profile follows the low-$\beta$ prescription shown in the top panel of Figure 2. The magnetic field increase toward the overshoot is less steep and the first peak is shifted further downstream.

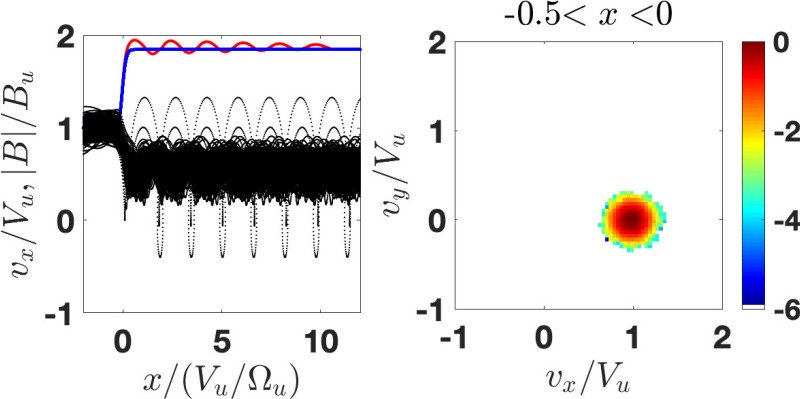

**Figure 6.** Left panel: the model (blue) and the predicted (red) magnetic profiles and the ion orbits $x$ vs $v_x$. Right panel: a slice of ion distribution inside the ramp. The shock parameters are $M = 1.9$, $\theta = 65°$, $R = 1.85$, $\beta_i = 0.05$, $\beta_e = 0.35$, and $s = 0.65$.

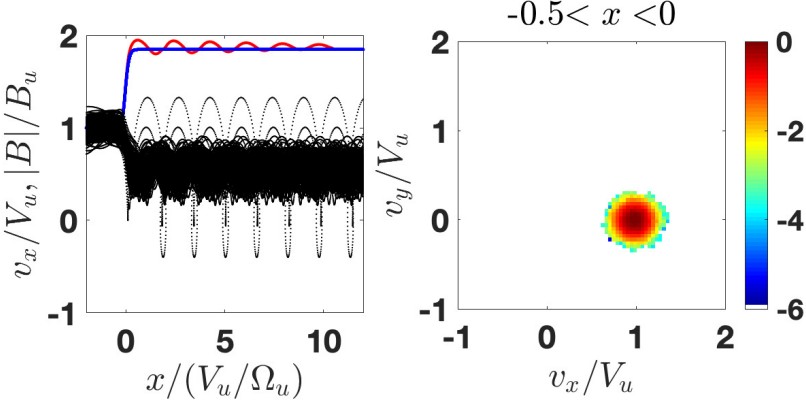

**Figure 7.** Left panel: the model (blue) and the predicted (red) magnetic profiles and the ion orbits $x$ vs $v_x$. Right panel: a slice of ion distribution inside the ramp. The shock parameters are $M = 2.05$, $\theta = 65°$, $R = 1.85$, $\beta_i = 0.2$, $\beta_e = 0.35$, and $s = 0.4$. The arrow points to the (quasi)reflected population.

Figure 8 shows the results of the test particle adjustment for the same compression ratio and $\beta_i = 0.4$ but lower cross-shock potential $s = 0.4$. This value is slightly above the value of $s_{5\%}$, so that the number of reflected ions is noticeable. Yet, the first

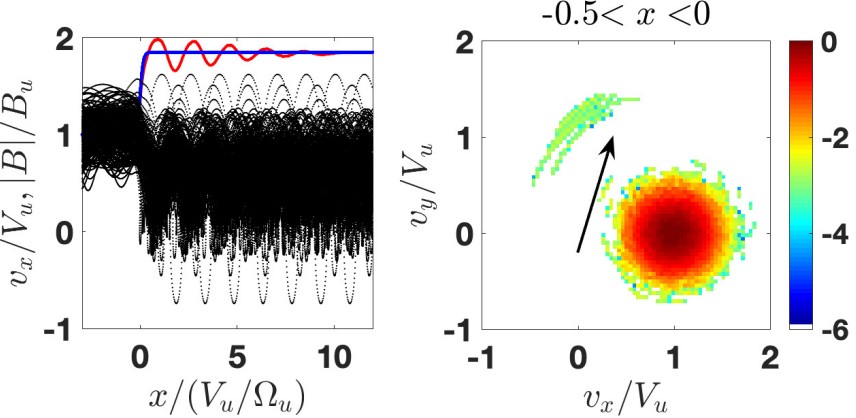

**Figure 8.** Left panel: the model (blue) and the predicted (red) magnetic profiles and the ion orbits $x$ vs $v_x$. Right panel: a slice of ion distribution inside the ramp. The shock parameters are $M = 2.1$, $\theta = 65°$, $R = 1.85$, $\beta_i = 0.4$, $\beta_e = 0.35$, and $s = 0.4$. The arrow points to the (quasi)reflected population.

maximum is shifted to downstream and the magnetic field increase toward the overshoot is not steep.

## 4 Observations

A detailed example of a pair of very low-Mach number shocks with the magnetic compression of $B_d/B_u \approx 1.2$ and $\beta_i \approx 0.08$ is given by Pope et al. (2019), Figure 4, where the cross-shock potentials are also calculated from observations and shown to agree well with the theoretical findings above. Namely, the shock with a lower potential has the first peak higher than the successive ones, while the shock with a higher potential has the second peak higher than the first one.

Figure 9 shows the magnetic profile of a of a subcritical shock observed by THEMIS B plotted over the ion spectrogram. The
shock crossing occurred at 2012/01/22 06:01:47. The estimated shock parameters are similar to those of Figure 5: $B_d/B_u = 1.85$, $\theta = 65°$, $\beta_i \approx 0.14$, and $M = 2.6$. The corresponding CGP is $s_{cr} \approx 0.71$ and CRP is $s_{5\%} \approx 0.64$. The spectrogram shows that a number of ions are quasi-reflected at the ramp. This is seen as a gap in the ion distribution inside the ramp. This gap cannot be seen using the standard "tplot" procedure of SPEDAS since the resolution is low. The IDL function "contour" makes an interpolation, similar to what is done when calculating distribution functions from a discrete set or measurements in a
number of energy channels and angle detectors. With this interpolation the gap becomes visible. Such quasi-reflection requires a sufficiently high cross-shock potential, capable of stopping slow ions inside the ramp (marked with a red arrow in the figure).

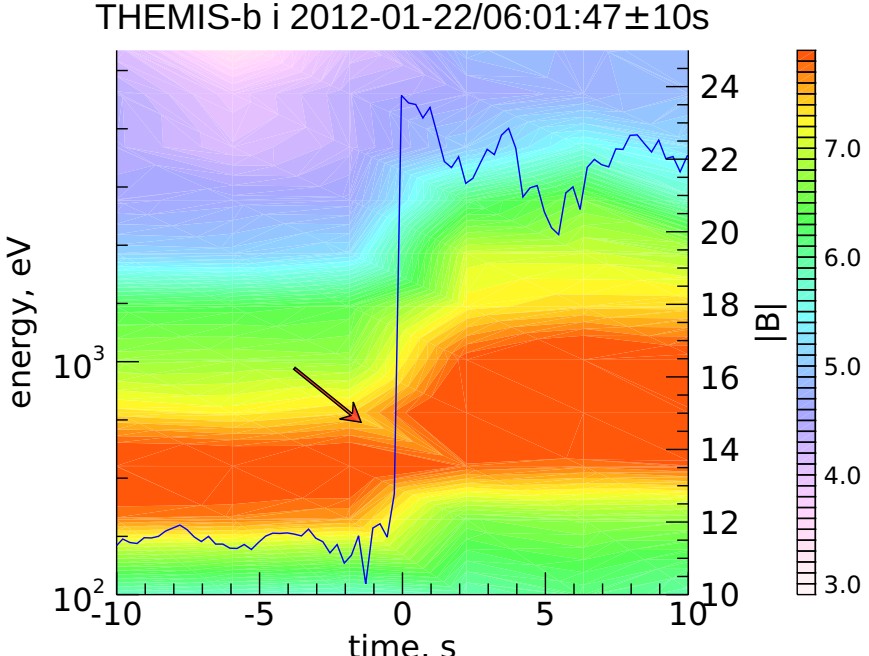

THEMIS-b i 2012-01-22/06:01:47±10s

**Figure 9.** Magnetic profile of 2012/01/22 THEMIS B measured shock.

The first peak follows a steep magnetic field increase and is the largest. Thus, we expect that $s_{5\%} < s < s_{cr}$, which is in a good agreement with the adjusted value of $s = 0.65$.

Figure 10 shows the corresponding gap for the analyzed shock in Figure 8. The spectrogram is made in the reference frame ("spacecraft") moving with the velocity $1.5V_u$ along the shock normal. Figure 11 shows a similar gap in 2011/11/28 THEMIS C measured shock spectrogram in which reflected ions are detected. It is not possible to compare directly the gap for the analyzed shock with observations since the analysis is done in the normal incidence frame while the observed spectrograms are produced in the spacecraft frame.

Figure 12 shows the magnetic profile of a THEMIS C observed shock. This shock is also subcritical. It has a lower magnetic compression $R = 1.4$ with a slightly higher $\beta_i \approx 0.2$. The angle is large $\theta = 86°$ while the Mach number is lower $M \approx 1.65$. The corresponding CGP is $s_{cr} \approx 0.5$ and CRP is $s_{5\%} \approx 0.38$. The absence of ions reflected inside the ramp indicates insufficient potential, so that we expect that $s < 0.38$. Adjustment using the advanced test particle analysis results in $s \approx 0.35$.

## 5 Discussion and conclusions

Magnetic field measurements at heliospheric shocks are by far the best quality measurements with regard to both precision and resolution. The resolution of particle measurements is much worse: their precision is limited by geometric factors and

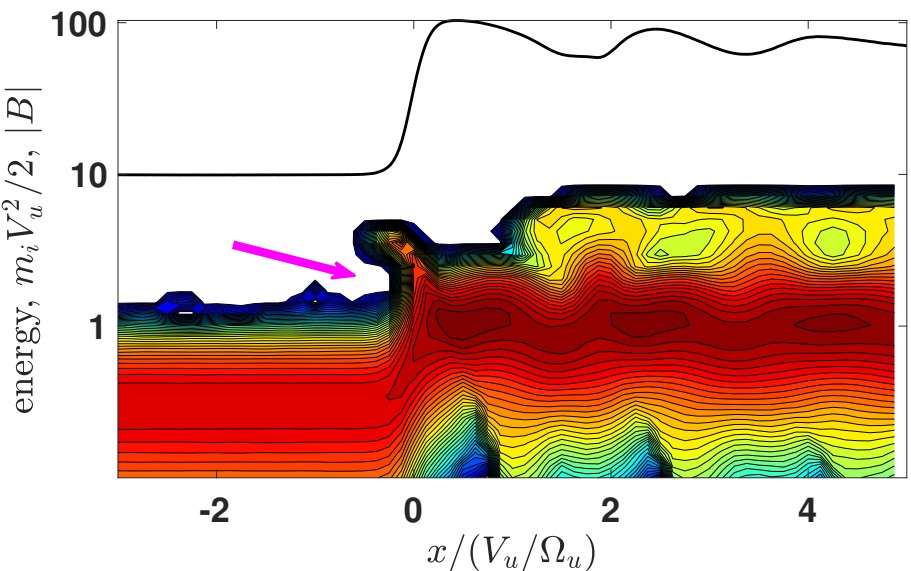

**Figure 10.** The gap for the shock in Figure 8.

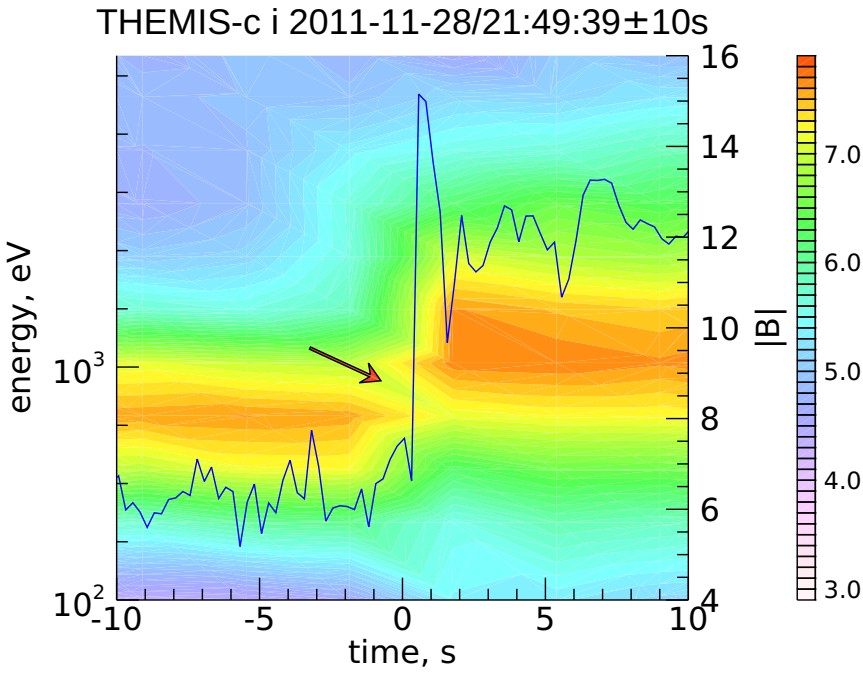

**Figure 11.** Magnetic profile of 2011/11/28 THEMIS C measured shock with the gap in spectrogram due to reflected ions.

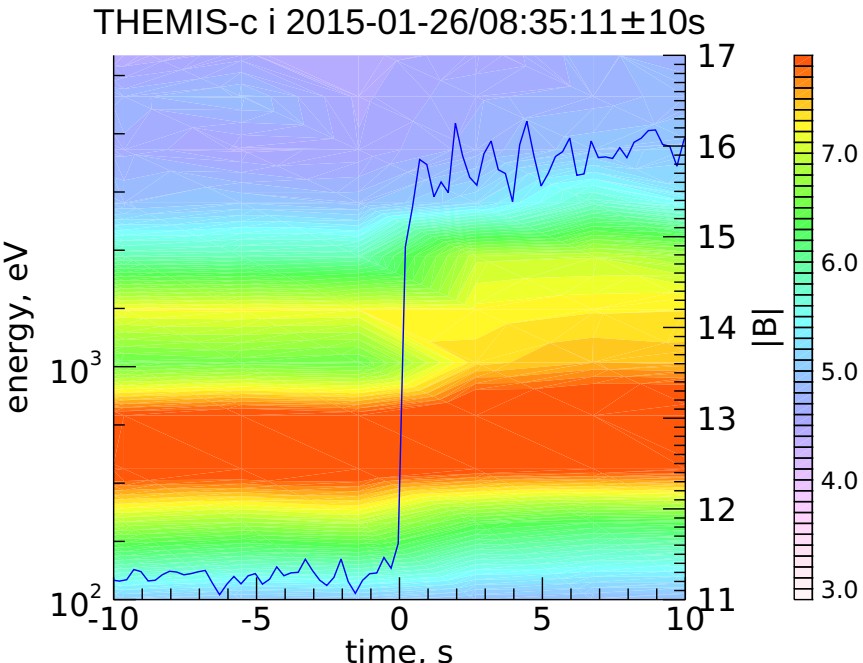

**Figure 12.** Magnetic profile of 2015/01/26 THEMIS C measured shock.

the finite number of detectors. Measurements of electric field are typically the most difficult ones. Therefore, any cross-check of less reliable measurements on the basis of better ones is important. In particular, if measurements of the magnetic field would enable us to fill gaps in particle and cross-shock measurements that would substantially improve our ability to compare observations and theory.

5    In the present paper we examine the implications of the shape of the downstream magnetic oscillation trail for the cross-shock potential. It appears that certain limitations can be placed on the potential using knowledge of the Mach number, magnetic compression, $\beta_i$, and first peaks of the downstream magnetic field. The two critical kinetic phenomena are the gyration of the center of the incident distribution upon crossing the shock and the onset of ion reflection within the ramp. These two features are related to the two critical values of the cross-shock potential that have been defined in the simplified case of a narrow perpendicular shock. The derived CGP $c_{cr} = 1 - (B_u/B_d)^2$ and CRP $c_{5\%} = (1 - 2v_T/V_u)^2$ are approximations which do not take into account properly the ramp width and the shock angle. Yet, they provide certain limits on possible cross-shock potentials consistent with the measured Mach number, $\beta_i$, and magnetic compression. Numerical test particle analyses have shown that these limits are in good agreement with the parameters obtained by adjustment of the predicted profile to the required downstream asymptotic value.

15    It is found that for $s_{cr} < s < s_{5\%}$ the first downstream peak is at the downstream edge of the ramp and is weaker than the second one. For $s < s_{cr} < s_{5\%}$ and for $s < s_{5\%} < s_{cr}$ the first downstream peak is shifted farther downstream and it is the

strongest. For $s_{5\%} < s < s_{cr}$ reflected ions are seen, the rise toward the overshoot is substantially steeper, the first downstream peak is at the downstream edge of the ramp and is the strongest. Thus, observations of the downstream magnetic oscillations may be used to place restrictions on the cross-shock potential. At this stage the analysis is limited to subcritical, marginally-critical and weakly supercritical shocks. Higher super-criticality will require separate study, including also post-ramp reflected ions.

*Competing interests.* No competing interests are present.

*Acknowledgements.* MG was supported in part by the Israel Science Foundation (grant No. 368/14). The work of Xiaoyan Zhou was supported by NASA NNX17AI26G.

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
