# Peer review of "Overshoot dependence on the cross-shock potential"

_Annales Geophysicae, 2019_

## Referee Comment (RC1) · Anonymous Referee #1 · 22 Sep 2019

This paper deals with the structure of the magnetic field (for instance the bow shock) downstream of the shock ramp. This structure depends on many parameters, in particular the Mach number (the shock being sub- or super-critical), plasma-$\beta$, width of shock ramp, shock potential and shock normal angle. All these parameters span a more-dimensional parameter space thus obscuring the relations between the structure of the magnetic field within a shock crossing. Most interesting in this structure is that super-critical shocks posses a magnetic overshoot while shocks at smaller Mach numbers, in particular sub-critical shocks exhibit an about coherent magnetic field oscillation across the shock ramp consisting of a few sequential maxima and minima the nature of which still remains unclear, while it seems that they should be produced by gyrating ions, partially maintaining their gyration phases when passing the shock.

[Figure]

The present paper investigates this interesting effect in order to pin down the nature of these coherent, so to say, oscillations in relation to the gyration of the shock-crossing ions under simplifying strictly quasi-perpendicular shock and low-$\beta$ conditions. Electrons in this investigation, which restricts itself to the ion motion, are not considered as they are of no interest here where the ions are treated as a cold beam whose dynamics inside the ramp suffers from the presence of the shock potential in which the ions drift along the shock and gyrate around the field. The authors attack the problem in what they call an advanced test particle model (they developed) with predescribed magnetic field shock-ramp profile and magnetic compression ratio and tracing the coherently gyrating ion beam across the shock when varying other parameters, of course assuming quasineutrality being warranted by the electrons and their pressure through a polytropic electron equation of state. Within this model they are able to identify the location and amplitudes of the magnetic field maxima resulting from the coherent ion gyration across the shock in dependence of the various varied parameters. Their interest focusses in particular on the effect of the cross-shock potential. The authors define two kinds of critical shock potentials, one related to gyration in the shock ramp for given shock ratio, the other related to the potential necessary to reflect the ions from the ramp. They find interesting relations between these, the downstream ion dynamics and the location and damping rates of the coherent magnetic downstream maxima. This contributes to the illumination of the structure of the magnetic field in the shock-adjacent downstream layer.

Results are given as plots of the oscillating structure of the ion beam of given velocity spread as function of distance across and behind the ramp. These are compared to THEMIS observations. Moreover ion reflection effects are also discussed giving rise to the identification of an energy gap in the ion distribution located at the shock ramp front. Various dependencies on the shock potential and the two critical potentials are discussed.

I find this a very useful examination of gyrating ion motion across low Mach number

about perpendicular shocks and their effect on the magnetic structure of the shock magnetic field downstream behind the shock. Even though it was previously suggested qulitatively that the magnetic oscillations observed in this domain would relate to the shock passing ions the direct demonstration of it was still missing. The present paper provides it which is an important contribution to the understanding of a particular class of shocks: quasi-perendicular low-Mach number shocks.

I suggest that the paper, which contains a large number of illuminating figures, is published essentially without any larger modifications as the theory is lucid, straightforward and concisely presented, well illustrated and accompanied by comparisons with real measurements in space which confirm it. Moreover, the discussion of the two critical potentials is an important turn which helps understanding the ion-magnetic field dynamics under the influence of a given cross shock potential.

There is a typo in the $<, >$ signs in line 5ff which should be corrected.

I find the inserts in Fig 1, and in Fig 3 the descriptions on the ordinates and abscissas and color bar about unreadable.

I think it is not necessary to give any more decimals than the first on the color bar.

In Fig2 the high and low should also be enlarged.

Numbers like 100 and 1000 etc should be given as $10^2$, $10^3$. Same in Fig 9, 11, 12.

These figures will all become one column figures. This requires that the lettering must be large enough for the reader to deciffre it.

---

## Referee Comment (RC2) · Anonymous Referee #1 · 17 Oct 2019

I am sorry for this long delay which, as I realize, is due to the laziness of the other reviewer.

I nevertheless strongly suggest not to withdraw the paper. I hope the editor-in-charge will handle the paper as quickly as possible, either urging the other reviewer to submit a review almost immediately or stay with my review only.

I would like to stress that I am satisfied with this paper and would not see any serious reason for not accepting it.

---

## Author Comment (AC1) · 17 Oct 2019

We are grateful to the referee for useful comments. All requested changes have been made long ago but we are now allowed to post a revised version until the editor makes a decision. At this stage it seems unlikely. The editor has not been able to move the review procedure forward by collecting a second review and we experience already a third extension until November 22 (almost 4 months since the submission date). I am also refused to know whether this extension is the last one of it will be done indefinitely. At present we are considering withdrawal. My apologies.
* * *

---

## Referee Comment (RC3) · Anonymous Referee #1 · 22 Nov 2019

Sorry, I cannot do anymore but alerting the EiC which I did.

I must admit that

1. the behaviour of reviewers who extend the response time till its very end or longer is scientifically not acceptable. It violates any serious scientific behaviour.

2. in addition I consider the entire discussion phase a nuisance. It artificially delays publication and is of no other value than letting the journal feel itself important. It contributes zero to science and thus is of no scientific value.

A simple check confirms that hardly any paper submitted to Angeo has been "discussed" in public, whatsoever this may mean. The only conversation is with the reviewers. For this no discussion phase is needed.

[Figure]

Moreover, the discussion phase pre-informs the interested public about ideas. This may become sensible as sometimes someone may be fast and publish quickly in another journal without discussion phase.

One should not believe that scientist are generous or honest people, by far not when it comes to the question of priority or ranking. Scientists are egocentric as almost everybody else, sometimes even more.

It is only the scientific method of correctness in experiment and logic which forces them to be scientifically honest, not in any other human sense.

Taking this into account the discussion phase is even counterproductive. There are sufficiently many cases in science where reviewers took their chance to benefit from reviewing a paper containing brandnew ideas, or picking up an idea otherwise in conversation or discussion. Such ideas are then said as having been in the air and have been developed independently at about the same time by different people or groups. It is symptomatic that in conferences and meetings people only very rarely present ideas which have not yet been submitted or are in print or even published.

---

## Short Comment (SC1) · 22 Nov 2019

Dear Editor,

This is completely unacceptable. Sometimes editors have to make tough decisions, this is what they are for. At this stage I am asking for a decision (whatever it will be) within two days. In you are not able to make one (which is effectively silent rejection) please consider the paper as withdrawn.

Sincerely yours, Michael Gedalin

Begin forwarded message:

From: <editorial@copernicus.org> Subject: angeo-2019-111 (author) - exten-

sion of discussion phase Date: 22 November 2019 at 14:26:06 GMT+2 To: <gedalin@exchange.bgu.ac.il>

Dear Michael Gedalin,

Please be aware that we had to extend the discussion of your following paper in AN-GEOD because additional Referee comments are needed:

Title: Overshoot dependence on the cross-shock potential Author(s): Michael Gedalin et al. MS No.: angeo-2019-111 MS Type: Regular paper

The new planned end date is 30 Nov 2019. You are invited to monitor the processing of your manuscript via your MS Overview: https://editor.copernicus.org/ANGEO/my_manuscript_overview

In case any questions arise, please do not hesitate to contact me.

Kind regards,

Natascha Töpfer Copernicus Publications Editorial Support editorial@copernicus.org

on behalf of the ANGEO Editorial Board

---

## Referee Comment (RC4) · Anonymous Referee #2 · 29 Nov 2019

This paper is a very interesting and elegant treatment of the magnetic overshoot structure of collisionless shocks including comparison of the presented analytical/numerical results with 2 periods of observation of Earth's quasi-perpendicular bow shock by THEMIS. The analysis is based on both analytical and numerical calculations of the proton orbits in a low-beta plasma at both perpendicular and quasi-perpendicular shocks for a variety of ramp potentials with the magnetic field determined by pressure balance. The analysis enables the ramp potential to be estimated based on the pattern of the downstream magnetic field magnitude oscillations.

In general I would recommend the paper for publication in Annales Geophysicae. However, I think the paper would benefit from some introductory text on collisionless shocks, and outstanding challenges in our understanding of them, in order that the paper is

more accessible, interesting and useful, for readers who are not experts in the subject.

Specific issues are:

1. (page 1, line 3) ….two critical cross-shock potentials are defined….

2. (1, 4) How is the "normal" velocity defined? Normal to what?

3. (1, 5) Plasma-beta should be defined.

4. (1, 16) Define "super-critical."

5. (2, 4) Sufficient attention has not been devoted…..

6. (2, 6) ….as well as about…..

7. (2, 15 – 20) The distinction between v and u is not made clear.

8. (3, 4) ….derive an analytical….

9. (3, 5) ….to weak…

10. (3, 12) ….velocity decreases…

11. (3, 18) ….field peak will….

12. (3, 22) ….and the spread…

13. (4, 26) ….not affect the…..

14. (6, 14) …..compression, CGP………... = 2, CGP is…..

15. (6, 21) …. There are two…..

16. (7, 1) …..but turns around….

17. (7, 11) …For an initial….

18. (7, 12) Insert a comma at the beginning of the line

19. (7, 16) …..point along the ion…..

20. (8, last line) . . .by the test-particle analysis. . ..

21. (9, 4) . . ..show the presence. . ..

22. (11, 6) . . .of a subcritical shock observed by THEMIS B plotted over. . ..

23. (11, 9) . . ...of ions are quasi-reflected. . ...

24. (11, 10) . . ..makes an interpolation, similar. . ...

25. (12, 2) . . ..shows a similar gap in a 2011/11/28. . ..

26. (12, 3) . . ...spectrogram in which reflected ions. . ..

27. (12, 8) . . ...The absence of ions reflected inside. . ...

28. (12, 11) . . ..quality measurements with regard to both precision. . ..

29. (12, 12) . . ..much worse: their precision. . ...and the finite number. . ...

30. (12, 14) . . ...would enable us to fill. . ..

31. (12, 15) . . .improve our ability to compare observations and theory.

32. (12, 17) . . ..examine the implications. . ...

33. (13, 2) . . ..and the onset. . ...These two features. . ...

34. (13, 3) . . ..potential that have. . .. . .

---

## Author Comment (AC2) · 30 Nov 2019

We are grateful to the referee for useful comments. All suggested changes have been implemented and the authors are waiting for permission to upload a revised version.

---

## Author Comment (AC3) · 30 Nov 2019

We are grateful to the reviewer for useful comments. Revisions are made in accordance with the comments and ready for submission. All text corrections are implemented. All parameters, which were not properly defined in the previous version, are defined now. The following introductory paragraph is added:

"Collisionless shocks (CS) are one of the most efficient accelerators of charged particles in the Universe. They are present in 15 virtually all plasma environments at the scales from âĹij 1 cm in the terrestrial labs to âĹij 1 Mpc in galaxy clusters. CS is a multi- scale object, where highest energies are achieved at largest scales within the diffusive process due to scattering at fluctuations far upstream and far downstream

and multiple crossings of the shock. The latter occur within the scatter-free region so that ion dynamics in the shock front is intimately related to the large scale acceleration: while the diffusive acceleration occurs at scales much larger than the shock width, the spectrum of the accelerated particles is essentially determined by conservation laws 20 at the scatter-free shock transition. The fields in the shock front are responsible for ion heating, generation of backstreaming ion beams (Burgess, 1987; Kucharek et al., 2004; Oka et al., 2005; Gedalin et al., 2008; Gedalin, 2016b), acceleration of pickup ions (Lee et al., 1996; Zank et al., 1996; Zilbersher and Gedalin, 1997; Ariad and Gedalin, 2013), and injection into the diffusive mechanism (Scholer et al., 2002; Giacalone, 2005). Thus, the structure of the shock front is the central problem of the shock physics. The shock structure can be studied within in situ measurements only at heliospheric shocks. Qualitative understanding of the shock structure substantially improved due to these high quality observations and also due to numerical simulations. The frontier of the observational shock studies has shifted recently towards the processes occurring within few ion convective gyroradii in both directions from the ramp along the shock normal (Dimmock et al., 2012; Wilson et al., 2012, 2014; Johlander et al., 2016; Burgess et al., 2016; Eselevich et al., 2017; Wilson III et al., 2017; Gingell et al., 2017). "

---

## Author Response (AR1)

[revised manuscript text omitted]

List of changes:

1. An introductory paragraph is added in the Introduction (in blue).

2. All figures are modified as requested.

3. All suggested textual changes are implemented.

5    4. All parameters are now defined. Most of the added definitions are in the beginning of section 2 (in blue).